# The Association Between Delivery of Small-for-Gestational-Age Neonate and Their Risk for Long-Term Neurological Morbidity

**DOI:** 10.3390/jcm9103199

**Published:** 2020-10-02

**Authors:** Omer Hadar, Eyal Sheiner, Tamar Wainstock

**Affiliations:** 1Faculty of Health Sciences, Joyce & Irving Goldman Medical School at Ben-Gurion University of the Negev, Beer-Sheva 8410501, Israel; 2Department of Obstetrics and Gynecology, Soroka University Medical Center, Ben-Gurion University of the Negev, Beer-Sheva 8410501, Israel; sheiner@bgu.ac.il; 3The Department of Public Health, Faculty of Health Sciences, Ben-Gurion University of the Negev, Beer-Sheva 8410501, Israel; wainstoc@bgu.ac.il

**Keywords:** small-for-gestational-age, neurological morbidities, developmental disorders, degenerative disorders

## Abstract

Small-for-gestational-age (SGA) is defined as a birth weight below the 10th or below the 5th percentile for a specific gestational age and sex. Previous studies have demonstrated an association between SGA neonates and long-term pediatric morbidity. In this research, we aim to evaluate the possible association between small-for-gestational-age (SGA) and long-term pediatric neurological morbidity. A population-based retrospective cohort analysis was performed, comparing the risk of long-term neurological morbidities in SGA and non-SGA newborns delivered between the years 1991 to 2014 at a single regional medical center. The neurological morbidities included hospitalizations as recorded in hospital records. Neurological hospitalization rate was significantly higher in the SGA group (3.7% vs. 3.1%, OR = 1.2, 95% CI 1.1–1.3, *p* < 0.001). A significant association was noted between neonates born SGA and developmental disorders (0.2% vs. 0.1%, OR = 2.5, 95% CI 1.7–3.8, *p* < 0.001). The Kaplan-Meier survival curve demonstrated a significantly higher cumulative incidence of neurological morbidity in the SGA group (log-rank *p* < 0.001). In the Cox proportional hazards model, which controlled for various Confounders, SGA was found to be an independent risk factor for long-term neurological morbidity (adjusted hazard ratio( HR) = 1.18, 95% CI 1.07–1.31, *p* < 0. 001). In conclusion, we found that SGA newborns are at an increased risk for long-term pediatric neurological morbidity.

## 1. Introduction

Small-for-gestational-age (SGA) is defined as birth weight below the 10th or below the 5th percentile or as two standard deviations below the mean birth weight for a specific gestational age and sex [1,2,3]. Dysfunction of the maternal innate immune response, genetic and chromosomal disorders, anemia, and malnutrition may all contribute to increased risk of being born as SGA [4,5]. Nevertheless, the determinants and the main pathophysiological pathways are not yet fully understood [4,5].

During the late 1980s, Barker was the first to demonstrate that SGA infants are at higher risk for long-term morbidity such as coronary heart disease [6,7]. The developmental origins of adult health and disease hypothesis considered early life as a critical period of development, in which optimal environment is critical for long term health and development. Following the Barker innovative theory, numerous studies have shown that SGA is associated with adverse physiological outcomes such as gastrointestinal disorders [2], chronic endocrine pathologies [8], and long-term pediatric ophthalmic morbidity [3].

Animal studies have demonstrated architectural alterations in the brains of the animal SGA models [9,10]. It has been shown that axonal and dendritic outgrowth was delayed, and reduced demyelination was evident [9].

Postmortem examination of human SGA brains shows a reduction in myelin lipids [11]. Interestingly, magnetic resonance imaging (MRI) studies of adolescents born as SGA were able to demonstrate a trend towards a smaller cerebral cortical, smaller frontal lobe, as well as smaller basal ganglia volume [12].

This body of evidence suggests that neonates born as SGA are at an increased risk for developing long-term neurological disorders. Indeed, SGA has been associated with an increased risk for neurocognitive disorders such as attention-hyperactivity symptoms and poor academic achievements [12,13]. However, other studies were not able to demonstrate that SGA had a significant impact on neurological morbidity [14].

Due to the controversy in the literature, it is important to understand whether SGA neonates are at increased risks for neurological morbidities. The current research aimed to study the possible association between being born as SGA and long-term neurological morbidity. 

## 2. Experimental Section

This population-based retrospective cohort analysis included all deliveries that took place at the Soroka University Medical Center (SUMC), the sole tertiary hospital in the Negev (southern Israel), between the years 1991 to 2014. Thus, the study is based on non-selective population data. The institutional review board (SUMC IRB committee) approved of the study in accordance with the ethical standards laid down in the 1964 Declaration of Helsinki (and its later amendments).

A comparison of the risk for childhood long-term neurological morbidity (up to the age of 18 years) was performed in offspring born as SGA and neonates who were diagnosed as appropriate -for-gestational-age (AGA). SGA in the current study was defined as birthweight <5th percentile for gestational age and gender, according to the WHO charts [15]. 

Multiple gestations, large for gestational age (>95th percentile) neonates and those with congenital malformations were excluded from the analysis. Data were collected from two separate databases that were cross-linked and merged for the study in order to match maternal data with offspring data; the SUMC (“Demog-ICD9”) computerized hospitalization database was used for the childhood data and the computerized perinatal database of the Obstetrics and Gynecology Department was used for the maternal and obstetric data. Demog-ICD9 database includes demographic information and ICD-9 codes for all medical diagnoses made during any hospitalization at SUMC. The perinatal database consists of information recorded immediately following delivery by an obstetrician. Experienced medical secretaries routinely review the information prior to entering it into the database to ensure its maximal completeness and accuracy. Coding is performed after assessing medical prenatal care records as well as routine hospital documents. Data regarding demographic characteristics, pregnancy, delivery, and adverse perinatal outcomes, as well long-term pediatric hospitalizations, were recorded and collected. First pediatric hospitalizations that included any neurological diagnoses were considered the outcome. The list of all neurological diagnoses by their ICD-9 is presented in Appendix A, and included the following subcategories: autism, eating disorders, sleep disorders, cerebral palsy (CP), headache, attention deficit hyperactivity disorder (ADHD), developmental disorders, and myopathy.

Follow-up of children was terminated if any of the following occurred: first hospitalization at SUMC with any of the listed morbidities (after which recurrent hospitalizations with the same diagnosis were not considered), when hospitalization resulted in death, end of study period, or when the child reached 18 years of age.

### 2.1. Statistical Analysis

Statistical analysis was performed using the statistical package for the social sciences (SPSS) 23rd ed. (SPSS, Chicago, IL, USA). Categorical data are shown in counts and percentages, and the differences were assessed by chi-square for the general association. The Student t-test was used to evaluate differences in continuous variables. A Kaplan-Meier survival curve was used to compare cumulative neurological morbidity incidences over time, and to evaluate the proportionality in risk over time between the study groups. The difference between the study groups was assessed using the log-rank test. 

A cox hazards model analysis was used to establish an independent association between SGA and future risk for offspring long-term neurological morbidity, while controlling for potential confounding variables based on the univariable analysis. The univariable analysis compared background characteristics of the study population between exposed and unexposed offspring, and between hospitalized and offspring not hospitalized. Additionally, maternal age was included in the model also due to its clinical significance.

These variables included: maternal age, gestational age at delivery, and hypertensive disorders. The possible co-linearity between the co-variables in the model was assessed using the Spearman correlation test, in which r > 0.6 (*p* < 0.05) was considered a strong correlation. All analyses were two-sided, and a p-value of < 0.05 was considered statistically significant.

## 3. Results

During the study period, 231,973 infants met the inclusion criteria; 10,998 (4.7%) were diagnosed as small-for-gestational-age neonates. maternal characteristics, morbidities and delivery outcomes for both groups are presented in Table 1. Mothers who had SGA newborns were younger (26.76 ± 5.880 vs. 28.11 ± 5.793; *p* < 0.001) and delivered at a later gestational age than the comparison group (39.20 ± 2.150 vs. 39.09 ± 1.978; *p* < 0.001). Low Apgar score at 5 min (5.5% vs. 2.1%; *p* < 0.001), and perinatal mortality (2.6% vs. 0.5%; *p* < 0.001) were significantly higher in the SGA group as compared to the non-SGA group.

Table 2 presents the incidence of selected neurological related hospitalizations of the offspring from birth until the age of 18 years in both study groups. The total neurological hospitalizations, (3.7% vs. 3.1%, OR = 1.2, 95% CI 1.1–1.3, *p* < 0.001) as well as development (0.2% vs. 0.1%, OR = 2.5, 95% CI 1.7–3.8, *p* < 0. 001) and demyelination disorders (0.13% vs. 0.06%, OR = 2.0, 95% CI 1.12–3.47, *p* = 0.007), were significantly higher in the SGA as compared to the non-SGA group.

In order to compare the long-term cumulative offspring neurological morbidity incidence over time, a Kaplan-Meier survival curve was constructed (Figure 1), showing a significantly higher cumulative incidence of offspring neurological morbidity in the small-for-gestational-age group.

A Cox multivariable regression model for the risk of offspring hospitalization with neurological morbidities is presented in Table 3. After controlling for maternal age and hypertensive disorders, SGA was found to be an independent risk factor for long-term neurological morbidity (adjusted HR = 1.18, 95% CI 1.07–1.31, *p* < 0.001).

## 4. Discussion

In this population-based study, children being born as SGA neonates were found to be at an increased risk for long-term neurological morbidities. Specifically, SGA newborns were at higher risk for having developmental disorders and degenerative illnesses. Their risk was higher even after adjusting for obstetrical confounding variables, including maternal age, diabetes mellitus, and hypertensive disorders. These findings reinforce previous knowledge that SGA neonates are found to have an overall increased neurological morbidity [16,17,18]. 

In this study, we aimed to expand the knowledge regarding neurological morbidity in SGA neonates. In order to do so, a wide range of morbidities were examined, such as neurocognitive disorders, neurodevelopmental disorders, and psychiatric disorders. 

Controversy exists regarding the nature of neurocognitive morbidity, which develops in SGA neonates. For example, some studies have shown neurocognitive deficits and decreased functional outcomes in SGA children, in particular attention deficit hyperactive disorder (ADHD) [12,18,19], while other studies did not find significant associations with SGA [14,20]. Our long-term follow up did not find an association between SGA and ADHD, but this may be explained by the fact that our study was hospital-based, and in most cases ADHD diagnosis is made in an outpatient setting.

Our results demonstrated a link between SGA neonates and that an increased risk of neurodegenerative and demyelination disorders is consistent with previous research. This finding co-interacts with MRI studies that show architectural alterations in the brains of adolescents [12] and with animal SGA models, which demonstrated reduced demyelination [9]. We suggest that the association between SGA neonates and demyelination disease may be a subject for future investigations, and further prospective studies are needed in order to understand the nature of those pathologies.

Our findings can be explained by the Barker’s hypothesis [21,22], describing that the uterine environment impacts the long-term health of the offspring. Evidence from other studies mange to show significant impact of the uterine environment and adverse physiological outcomes such as increased rates of cardiovascular disease [6,7], gastrointestinal disorders [2], chronic endocrine pathologies [8], and long-term pediatric ophthalmic morbidity [3]. A developmental model for the origins of disease suggests that nutrition during fetal life may affect fetal programing and its predisposition for disease may set the body’s organs and systems for future growth [23]. We suggest that this biological mechanism also impacts future central nerve system (CNS) development and may be associated with the increased neurological morbidity [24].

Numerous studies demonstrate that neurological morbidity such as neurodevelopmental disorders may play an important role in our society. Adverse economic outcomes such as longer hospitalization rate and greater median hospital cost have found to be related with children diagnosed with neurological disorders [25]. Moreover, multiple studies have demonstrated that neonates born with neurodevelopmental disorders are at increased risk for future emotional and adaptive impairments [26,27]. Our study emphasizes the connection between being born as SGA neonate and future risk for neurological morbidity. We suggest that careful surveillance and early diagnosis in children with such disorders may carry a critical effect on later offspring health and perhaps improve their prognosis.

In our study we used actual birth weight rather than IUGR, which is based on estimated weight. Unfortunately, our database does not include data on Doppler studies which can predict future growth restricted neonates, who carry a higher risk for adverse perinatal outcomes [28].

The main strength of our study is the large, population-based data. SUMC is the only tertiary medical center in the South, providing both obstetrical services as well as neurological services. This results in a great advantage, as it allows us to follow-up on neonates born at our medical center throughout their childhood and into adulthood.

Nevertheless, our sizeable population-based retrospective cohort analysis has also some limitations. The retrospective nature of the research restricts our ability to assume causality, and solely allows us to form proof of association. In addition, patients who moved out of the area were lost to follow up examinations. Nonetheless, we can assume that such withdrawals were relatively equal in both the study groups.

Significant limitations that we should consider are carefully related to the nature of the neurological and neurocognitive conditions included in the study. Some neurological conditions, such as learning disabilities, are mostly treated in an outpatient setting and were not or under-represented in our cohort. In comparison, conditions that are mostly treated in SUMC, such as neurodegenerative disorders, show overall increased morbidity in SGA neonates.

## 5. Conclusion

We conclude that SGA newborns appear to be at an increased risk for pediatric neurological morbidity. Since the mechanism by which SGA neonates develop neurological morbidity is not fully understood, further studies are recommended.

## Figures and Tables

**Figure 1 jcm-09-03199-f001:**
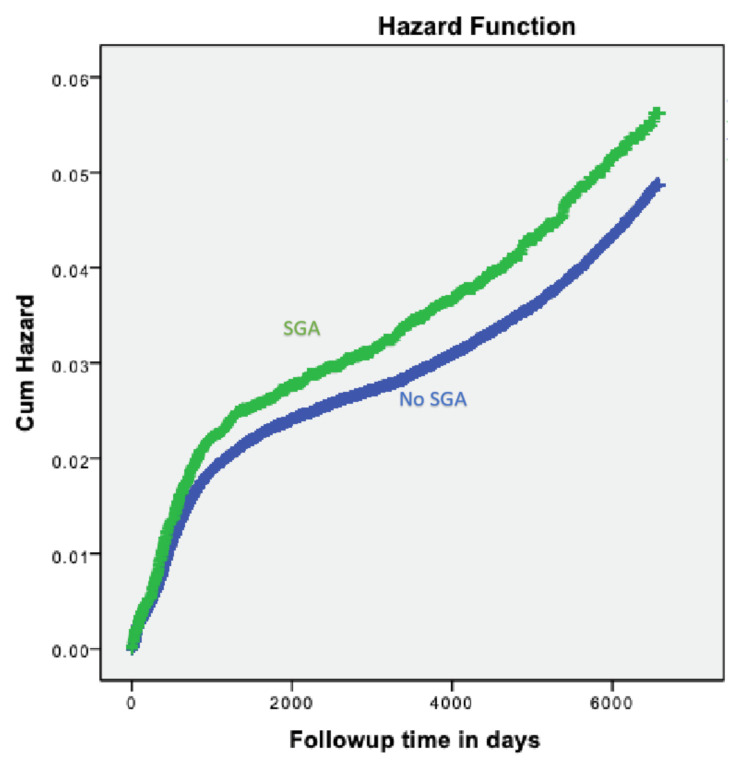
Kaplan–Meier survival curve demonstrating the cumulative incidence of total hospitalizations with neurological morbidities in SGA vs. non-SGA neonates (log-rank *p* < 0.001).

**Table 1 jcm-09-03199-t001:** Maternal characteristics and delivery outcomes based on study group.

*p*-Value	SGA Newborns (N = 11,290) % (n)	Not SGA Newborns (N = 221,978) % (n)	Maternal and Newborn Characteristic
<0.001	26.76 ± 5.880	28.11 ± 5.793	Maternal age (years, mean ± SD)
<0.001	39.20 ± 2.150	39.09 ± 1.978	Gestational age (weeks, mean ± SD)
<0.001	2399.95 ± 435.724	3201.58 ± 371.473	Birth weight (grams, mean ± SD)
Parity (%)			
<0.001	37.7 (4258)	23.5 (52,095)	1
	43.2 (4879)	51.6 (114,574)	2–4
	19.1 (2151)	24.9 (55,259)	5+
0.13	6.6 (743)	7.0 (15,439)	Preterm (<37)
<0.001	2.4 (274)	4.6 (10,300)	Diabetes mellitus ^1^
<0.001	8.7 (986)	4.7 (10,591)	Hypertensive disorders ^2^
<0.001	52.0 (5870)	4.7 (10,487)	Low birth weight (≤2500 g)
<0.001	2.9 (327)	0.5 (1115)	Very Low Birth Weight Group (<1500 g)
<0.001	5.5 (618)	2.1 (4723)	Apgar scores <7 at 5 min (%)
<0.001	2.6 (292)	0.5 (1003)	Perinatal mortality

^1^ Including pregestational or gestational diabetes. ^2^ Including chronic hypertension, gestational hypertension, and preeclampsia.

**Table 2 jcm-09-03199-t002:** Univariate analysis comparing the incidence of Long-term neurological disorders in children (up to the age of 18 years) born as SGA or non-SGA.

*p-V*alue	OR; 95% Confidence Interval	SGA N = 10,998 N (%)	No SGA N = 220,975 N (%)	Neurological Outcome
0.408	1.827 (0.43–7.76)	2 (0.02)	22 (0.01)	Autism
0.511	0.849 (0.52–1.38)	17 (0.13)	402 (0.18)	Eating disorders
0.057	2.393 (0.94–6.05)	5 (0.04)	42 (0.018)	Sleep disorders
0.497	1.094 (0.84–1.42)	60 (0.54)	1102 (0.50)	Psychiatric emotional
0.901	1.050 (0.49–2.24)	7 (0.06)	134 (0.06)	ADHD
<0.001	2.527 (1.68–3.82)	26 (0.24)	207 (0.09)	Developmental disorders
0.007	2.048 (1.12–3.47)	15 (0.13)	148 (0.06)	Degenerative, Demyelination
0.049	1.843 (0.99–3.42)	11 (0.10)	120 (0.05)	Myopathy
<0.001	1.204 (1.09–1.33)	406 (3.71)	6820 (3.10)	Total neurologic hospitalizations

**Table 3 jcm-09-03199-t003:** Cox multivariable regression model for the risk of long-term neurological morbidity of the offspring.

Variables	Adjusted HR	95% CI		*p-*Value
		Min	Max	
SGA (vs. AGA)	1.18	1.07–1.31		<0.001
Maternal age (years)	0.997	0.99–1.00		0.116
Hypertensive disorders ^1^	1.130	1.03–1.24		0.018
Gestational age (weeks)	0.930	0.93–0.94		<0.001

^1^ Including chronic hypertension, gestational hypertension, and preeclampsia.

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
