# Peer review of "The Association Between Delivery of Small-for-Gestational-Age Neonate and Their Risk for Long-Term Neurological Morbidity"

_jcm, 2020, doi:10.3390/jcm9103199_

Round 1
Reviewer 1 Report
Most of the statistical methods employed in this manuscript are appropriate. But I have two concerns about the Cox Proportional Hazard Model:
(1) the Kaplan-Meier curve in Figure 1 suggests the proportional assumption might not hold. Have the authors check it?
(2) In the multivariate Cox regression, how are the covariates selected? by which criterion?
(3) check the proportional assumption before they fit the Cox
Proportional Hazard Model;
(4) give the criterion for covariate selection for the multi-covariate
regression. Frankly to say, both of them are very constructive
statistical suggestions. If these suggestions could be followed,
I think the statistics of manuscript would be improved and in a
great shape.
Reviewer 2 Report
the Manuscript is interesting nevertheless I need major revision.
no obstetrical data on Doppler flowmetry are described in the study, I guess you are not using the term -_SGA according delphi definitioon of 2016
please modify the paper
Author Response
Reviewer 2
the Manuscript is interesting nevertheless I need major revision.
no obstetrical data on Doppler flowmetry are described in the study, I guess you are not using the term -_SGA according delphi definitioon of 2016
please modify the paper
Dear Reviewer.
Thank you very much for your valuable suggestions.
Please, find the list of responses to your comments. We have addressed the comments below in this letter, and in the revised manuscript attached.
We sincerely hope the revisions we made will meet your approval.
- Following the comment: no obstetrical data on Doppler flowmetry are described in the study, I guess you are not using the term -_SGA according delphi definition of 2016
SGA in our study was defined as birth weight below the 5th percentile for a specific gestational age and sex. As we wrote in the Introduction: “Small-for-gestational-age (SGA) is defined as birth weight below the 10th or below the 5th percentile for a specific gestational age and sex” (lines 35-37). Meaning that in this study the actual birth weight was collected and not IUGR, based on the suspected fetal weight. Unfortunately, we do not have data regarding Doppler studies in our database. Thus we decided to concentrate on the actual small babies and not IUGR, using the Delphi definition.
We have added this important point to the discussion section, as a limitation of our study. We added to the Discussion section, that the Delphi procedure may detect future growth restricted neonates who carries higher risk for adverse perinatal outcomes (lines 215-217).
“In our study we used actual birth weight rather than IUGR which is based on estimated weight. Unfortunately, our database does not include data on Doppler studies which can predict future growth restricted neonates who carries higher risk for adverse perinatal outcomes [28].”
- Following the comment - please modify the paper
We added in the Introduction section a further explanation regarding Barker theory [ lines 43-46].
During the late 1980s, Barker was the first to demonstrate that SGA infants are at higher risk for long-term morbidity such as coronary heart disease [6]. The developmental origins of adult health and disease hypothesis, considered early life as a critical period of development, in which optimal environment is critical for long term health and development. Following Barker innovative theory, numerous studies have shown that SGA is associated with adverse physiological outcomes such as gastrointestinal disorders [2], chronic endocrine pathologies [8], and long-term pediatric ophthalmic morbidity [3].
Moreover, we added in the Discussion section a paragraph which discusses future neurodevelopment impact related to SGA neonates [lines 203-210].
“Numerous studies demonstrate that neurological morbidity has a significant influence on our society. Adverse economic outcomes such as longer hospitalization rate and greater median hospital cost have found to be related with children diagnosed with neurological disorders [25]. Moreover, neonates born with neurodevelopmental disorders are at increased risk for future emotional and adaptive impairments [26,27]. Our study emphasis the association between being born as SGA neonate and future risk for neurological morbidity. We suggest that carful surveillance and early diagnosis in children with such disorders, may carry a critical effect on later offspring health and perhaps improving their prognosis.”
In addition, we added several new references –
- Moreau JF, Fink EL, Hartman ME, Angus DC, Bell MJ, Linde-Zwirble WT, et al. Hospitalizations of children with neurological disorders in the United States. Pediatric critical care medicine : a journal of the Society of Critical Care Medicine and the World Federation of Pediatric Intensive and Critical Care Societies 2013 Oct 1,;14(8):801-810.
- Gardon L, Picciolini O, Squarza C, Frigerio A, Giannì ML, Gangi S, et al. Neurodevelopmental outcome and adaptive behaviour in extremely low birth weight infants at 2 years of corrected age. Early human development 2019;128:81-85.
- Aarnoudse-Moens CSH, Weisglas-Kuperus N, van Goudoever JB, Oosterlaan J. Meta-analysis of neurobehavioural outcomes in very preterms and/or very low birth weight children. Pediatrics. 2009 Aug;124(2):717-729. https://doi.org/10.1542/peds.2008-2816
- Gordijn SJ, Beune IM, Thilaganathan B, Papageorghiou A, Baschat AA, Baker PN, et al. Consensus definition of fetal growth restriction: a Delphi procedure. Ultrasound in obstetrics & gynecology 2016 Sep;48(3):333-339.
We sincerely hope the revisions we made will meet your approval.

Reviewer 3 Report
In this retrospective paper,
authors evaluated the association between small for gestational age children and long term risk for neurological morbidity.
I would suggest to discuss the significance and the consequences of the neurological disorders for our society. Additionally, you could also further extend the discussion about the thrifty phenotype theory!
Author Response
Reviewer 3
In this retrospective paper,
authors evaluated the association between small for gestational age children and long term risk for neurological morbidity.
I would suggest to discuss the significance and the consequences of the neurological disorders for our society. Additionally, you could also further extend the discussion about the thrifty phenotype theory!
Dear, Reviewer.
Thank you very much for your and valuable suggestions. Please, find the list of responses to your comment. we appreciate your remarks. We have addressed the comments below in this letter, and in the revised manuscript attached.
We sincerely hope the revisions we made will meet your approval.
Following the comment: I would suggest to discuss the significance and the consequences of the neurological disorders for our society.
- We added an additional paragraph which discusses consequences of the neurological disorders for our society, both in the economical level as well as in the social- development level. The following lines (203-210) have been added to the Discussion
“Numerous studies demonstrate that neurological morbidity has a significant influence on our society. Adverse economic outcomes such as longer hospitalization stay and greater median hospital cost have been found to be associated with children diagnosed with neurological disorders [25]. Moreover, neonates born with neurodevelopmental disorders are at increased risk for future emotional and adaptive impairments [26,27]. Our study emphasis the association between being born as SGA neonate and future risk for neurological morbidity. We suggest carful surveillance and early diagnosis in children with such disorders, may carry a critical effect on later offspring health and perhaps improving their prognosis.
“
Following the comment: Additionally, you could also further extend the discussion about the thrifty phenotype theory.
- We emphasis the paragraph which discuss Barker theory. We have added additional morbidities which related to future pathological outcomes of SGA neonates regardless to neurological morbidity. The following lines 195 - 198 has been added to the Discussion.
“Our findings can be explained by the Barker's hypothesis [21,22], suggesting that the uterine environment impacts the long-term health of the offspring. Evidence from other studies manage to show a significant impact of the uterine environment and adverse physiological outcomes such as increased rates of cardiovascular disease [6,7], gastrointestinal disorders [2], chronic endocrine pathologies [8], and long-term pediatric ophthalmic morbidity [3]. A developmental model for the origins of disease suggests that nutrition during fetal life may affect fetal programing and its predisposition for disease. Thus, it may set the body's organs and systems for future growth [23]. This biological mechanism may also impact future central nerve system development and may be associated with the increased risk for long-term neurological morbidity [24].”
In addition, we have added to the Introduction section sentence paragraph which emphasis Barker’s theory and its impact. Lines 43-46.
“During the late 1980s, Barker was the first to demonstrate that SGA infants are at a higher risk for long-term morbidity such as coronary heart disease [6, 7]. The developmental origins of adult health and disease theory, considered early life as a critical period of development, in which optimal environment is critical for long-term health and development. Following Barker innovative theory, numerous studies have shown that SGA is associated with adverse physiological outcomes such as, gastrointestinal disorders [2], chronic endocrine pathologies [8], and long-term pediatric ophthalmic morbidity [3]”
In addition, we added several new references –
- Moreau JF, Fink EL, Hartman ME, Angus DC, Bell MJ, Linde-Zwirble WT, et al. Hospitalizations of children with neurological disorders in the United States. Pediatric critical care medicine : a journal of the Society of Critical Care Medicine and the World Federation of Pediatric Intensive and Critical Care Societies 2013 Oct 1,;14(8):801-810.
- Gardon L, Picciolini O, Squarza C, Frigerio A, Giannì ML, Gangi S, et al. Neurodevelopmental outcome and adaptive behaviour in extremely low birth weight infants at 2 years of corrected age. Early human development 2019;128:81-85.
- Aarnoudse-Moens CSH, Weisglas-Kuperus N, van Goudoever JB, Oosterlaan J. Meta-analysis of neurobehavioural outcomes in very preterms and/or very low birth weight children. Pediatrics. 2009 Aug;124(2):717-729. https://doi.org/10.1542/peds.2008-2816
- Gordijn SJ, Beune IM, Thilaganathan B, Papageorghiou A, Baschat AA, Baker PN, et al. Consensus definition of fetal growth restriction: a Delphi procedure. Ultrasound in obstetrics & gynecology 2016 Sep;48(3):333-339.
Thank you very much for your interest in our study and your complements regarding its content. We sincerely hope the revisions we made will meet your approval.

Round 2
Reviewer 1 Report
For my first comments, checking the proportional assumption prior to fitting the Cox Proportional Hazard Model, it is a routine statistical requirement and multiple statistical testing methods, rather than observing the curves with eyeballs, are available, such as test based on Schonefeld residuals.
For my second question, confounding factor caused collinearity is a common statistical concern when fitting multivariate model. Model selection method, such as Bayesian Information Criterion, is usually helpful.
Reviewer 2 Report
thank you for your Rely,
the Manuscript improved
Author Response
Thank you very much for your and valuable comments.